# Brain Network Modeling Based on Mutual Information and Graph Theory for Predicting the Connection Mechanism in the Progression of Alzheimer’s Disease

**DOI:** 10.3390/e21030300

**Published:** 2019-03-20

**Authors:** Shuaizong Si, Bin Wang, Xiao Liu, Chong Yu, Chao Ding, Hai Zhao

**Affiliations:** School of Computer Science and Engineering, Northeastern University, Shenyang 110169, China

**Keywords:** Alzheimer’s disease, graph theory, mutual information, network model, connection mechanism, functional magnetic resonance imaging, topological structures, anatomical distance

## Abstract

Alzheimer’s disease (AD) is a progressive disease that causes problems of cognitive and memory functions decline. Patients with AD usually lose their ability to manage their daily life. Exploring the progression of the brain from normal controls (NC) to AD is an essential part of human research. Although connection changes have been found in the progression, the connection mechanism that drives these changes remains incompletely understood. The purpose of this study is to explore the connection changes in brain networks in the process from NC to AD, and uncovers the underlying connection mechanism that shapes the topologies of AD brain networks. In particular, we propose a mutual information brain network model (MINM) from the perspective of graph theory to achieve our aim. MINM concerns the question of estimating the connection probability between two cortical regions with the consideration of both the mutual information of their observed network topologies and their Euclidean distance in anatomical space. In addition, MINM considers establishing and deleting connections, simultaneously, during the networks modeling from the stage of NC to AD. Experiments show that MINM is sufficient to capture an impressive range of topological properties of real brain networks such as characteristic path length, network efficiency, and transitivity, and it also provides an excellent fit to the real brain networks in degree distribution compared to experiential models. Thus, we anticipate that MINM may explain the connection mechanism for the formation of the brain network organization in AD patients.

## 1. Introduction

Alzheimer’s disease (AD) is the primary form of dementia and the most common degenerative brain disease among older people [1]. AD patients show symptoms of a decline in memory, language, problem-solving and other cognitive functions that affects a person’s ability to perform daily activities. This decline occurs because the functional connections between two brain regions involved in cognitive function have been damaged, which blocks the normal information transformation [2,3,4]. Although a large amount of research has been devoted to Alzheimer’s disease, it is still a significant challenge to discover the underlying connection patterns that cause the alteration of functions in brain network of AD [5,6,7]. Addressing these problems have profound significance for a better understanding of how the interactions alter among brain regions, and it is crucial for the early detections and early treatments of AD.

More recently, both graph theory and mutual information (MI) have emerged as powerful and efficient mathematical techniques for investigating AD. Indeed, graph theory provides many topological properties for evaluating the characteristics of human brain networks [8,9,10,11,12]. Taking the form of a graph, people can learn the alterations of brain network properties in terms of network connectivity, transitivity, efficiency, degree distribution, modularity, and small-world-ness between normal controls (NC) and AD patients [13,14,15]. Particularly, graph theory also provides numerous methods of network modeling for simulating the evolution processes of real complex networks [16,17,18,19]. Through network modeling, one can surmise the fundamental causes that result in the existence of connections among nodes, and explain the underlying mechanisms of networks organization [20]. Previous studies have reported that network modelings could be effectively applied in various science fields to help explore the dynamic connection schemes in networks of real-world systems [21,22,23,24,25], e.g., friendships recommendation in social networks [21,22], and spurious links identification in biological networks [23,24]. We can generate network topologies that incorporate desired properties by employing suitable network models. Network modeling is viewed as a promising way to help us understand how the inter-connection mechanism affects the topological structures in complex networks.

It is worth mentioning that network modeling has made some significant advances, particularly in the study of brain networks simulation, with the intersecting developments of graph theory and network neuroscience [26,27,28,29,30,31]. One of the critical contributions that should be mentioned, for instance, is the Economical Clustering Model (ECM) proposed in [26]. ECM can construct the brain network by adopting the local topologies of common neighbors (CN) between two regions, and the authors tried to uncover the connection mechanism of the formation of brain networks. Simulation results show that networks modeled by ECM can dramatically capture a range of topological features of real functional brain networks. Similarly, Betzel et al. proposed a series of generative models of human brain networks according to different network structures. They aimed to explore the wiring rules that shape the topologies of the brain connectome [27]. Their efforts play a vital role in understanding how the cognitive function changes across the lifespan. Previous studies indicate that network modeling has emerged as an essential way in brain network investigations. By studying the associated connection mechanisms of the network models, we can explain the workings of systems built upon those networks.

Besides, as a fundamental quantity of information theory, MI provides a measure of the statistical dependence between two random variables [32,33]. At present, MI has been intensively investigated in the evaluation of the functional brain connectivity and the changes in interactions between mild cognitive impairment (MCI) and AD [34]. (Mild cognitive impairment (MCI) causes a slight but noticeable and measurable decline in cognitive abilities, including memory and thinking skills. A person with MCI is at an increased risk of progressing Alzheimer’s or another dementia [35]). Besides, MI is applied to estimate the probability of successfully information transmission over the brain connections between different cortical regions of patients suffering from AD [36]. Moreover, previous studies have found that MI can be used to quantify the effect of correlations between the Mini-mental state examination (MMSE) scores of AD and cognitive stage [37,38]. (Mini-mental state examination (MMSE) is a method to evaluate the cognitive state of AD patients and has been routinely used in clinical settings [39]). By comparing the MMSE scores between MCI and the moderate stages of AD, researchers find that the cognitive ability of probable AD patients declines more severe than that of MCI [40]. Furthermore, MI is treated as an effective measure in evaluating the inter-relationships of gene expression networks for AD and other neurodegenerative diseases [41,42]. Consequently, previous studies indicate that MI has a significant influence in brain research of AD.

However, it is worth mentioning that mutual information between different cortical regions in brain networks has not been considered in previous brain network models. Additionally, although extensive research has been carried out on brain network modeling, far too little attention has been paid to the dynamic process of brain networks simulation from one existing state, e.g., NC, to another, e.g., AD. Most of the previously proposed models start their simulation with the assumptions that nodes are isolated and no connections exist in the initial simulation networks. Apparently, these models are not suitable for simulating the brain networks changes from NC to AD. Moreover, these models generate the desired synthetic networks by continually adding connections into the initial simulation networks. They have ignored the fact that real brain networks combine both emerging new connections and disappearing old connections in the progression from NC to AD [43,44]. Therefore, models that only take adding connections into account are not realistic. To realize a better brain networks construction of AD patients, the models should consider adding and deleting connections, simultaneously.

Considering the above, in this work, we propose a network model named MINM for brain network modeling. Both MI and graph theory are used to simulate the connection changes in the progression of human brain networks of AD. The ultimate goal of the study is to uncover the connection mechanism that produces synthetic networks with properties similar to those of real observed brain networks topologies. The rest of this paper is organized as follows. In Section 2, we introduce the materials and methods, including the construction of real brain network, the presentation of our proposed models MINM in detail and the evaluation of synthetic networks. Then, we show the results of our experiments in Section 3. Section 4 is devoted to performance analysis and discussion of our proposed MINM. Finally, in Section 5, we draw the conclusion of this work.

## 2. Materials and Methods

### 2.1. Data Acquisition and Participants Selection

Data used in this study were recruited from the public resting-state functional magnetic resonance imaging (rs-fMRI) datasets named Alzheimer’s Disease Neuroimaging Initiative (ADNI) (http://adni.loni.ucla.edu) consisting of a total of 147 participants. They are divided into three groups: normal controls (NC) group, mild cognitive impairment (MCI) group, and Alzheimer’s disease (AD) group. Table 1 shows the demographic and clinical characteristics of the three groups. The NC participants were non-depressed, non-demented, and had an average MMSE score of 28.72. The MCI group had an average MMSE score of 27.68. Patients with AD had an average MMSE score of 22.36. Each participant underwent a scan session using a 3.0T Philips MRI scanner. All the resting fMRI scans were collected axially by adopting an echo-planar imaging (EPI) sequence with the following parameters: repetition time (TR) = 3000 ms; echo time (TE) = 30 ms; axial slices = 48; slice thickness = 3.313 mm; slice acquisition order = sequential ascending; and flip angle (FA) = $80.0^{\circ }$. Participants were informed to relax their minds and keep their eyes closed during the scanning to obtain resting state MRIs.

### 2.2. Data Preprocessing

A standard data preprocessing strategy was performed in our study using the Data Processing Assistant for Resting-State fMRI (DPARSF) software (http://www.rfmri.org/DPARSF) [45] and the well-known Statistical Parametric Mapping software package (SPM8) (http://www.fil.ion.ucl.ac.uk/spm) [46]. The data preprocessing for each resting-state scan contained the following steps: (1) To guarantee the stabilization of the magnetic field, the first ten slice time points were discarded. (2) **Slice timing correction** was performed to ensure all remaining time points in the correct time domain. (3) **Realignment** was then executed to eliminate the movement artifact in the BOLD time series. Participants whose head translation exceeded 3.0 mm and participants whose head rotated more than $3.0^{\circ }$ were discarded. (4) The functional volumes would subsequently be **Spatial normalized** to the standard EPI template and re-sliced to 3×3×3 mm${}^{3}$ resolution in Montreal Neurological Institute (MNI) space. (5) **Spatially smoothing** was further performed on the normalized images using a Gaussian kernel of 4 mm full width at half-maximum (FWHM). (6) To reduce the influence of low-frequency drifts and high-frequency noise, temporal band-pass **filtering** in the frequency range 0.06–0.11 Hz was achieved over each smoothed images. (7) Both linear and quadratic trends were removed. (8) Nuisance covariates such as six head motion parameters, whole-brain signal, cerebrospinal fluid, and white matter were regressed out from the preprocessed data.

### 2.3. Construction of Real Brain Network

The real brain network of each participant is represented by a binary graph in this work. First, we used the automated anatomical labeling (AAL) template [47], which functionally parcels the cerebrum into 90 regions (45 regions for each hemisphere) of interest (ROIs), to define the nodes in the graph. Second, we calculated the Pearson correlation coefficient between the whole-run BOLD time courses of any pairs of nodes, to obtain one 90 × 90 inter-regional symmetric correlation matrix for each participant. Then, Fisher’s r-to-z transformation was performed to improve the normality of the correlation coefficients in the matrix. Third, each inter-regional correlation matrix was threshold to retain a fraction of the strongest connections for statistical significance. The element in the matrix was set to 1, with the condition that the corresponding correlation coefficient of the node pair was greater than a given threshold θ. Alternatively, it was set to 0. Finally, we obtained a binary graph *g* to represent the real brain network of one participant. The element g(u,v)=1 means that there is a functional connection between node *u* and node *v*. Moreover, different connectivity densities of the real brain networks were generated by defining different θ.

### 2.4. Synthetic Brain Network Modeling

In this study, we aimed to simulate the progression from NC to AD through network modeling. Here, we call the networks generated by our proposed model synthetic brain networks, and the networks constructed from the preprocessed images real brain networks. By comparing the properties of synthetic networks with real target brain networks, we could predict the connection mechanism that causes the topological alterations of AD.

#### 2.4.1. Network Modeling Steps

The dynamic process of synthetic brain network modeling was executed step by step, starting from the **Initialization** step, followed by the **Connection probabilities calculation** step and the **Evolution** step. The detailed network modeling steps are listed in the following:Initialization: At the beginning of modeling, all participants in groups of NC, MCI and AD were preprocessed and the corresponding inter-regional correlation matrices $G_{n}=\left(g_{n1},g_{n2},\ldots ,g_{nk_{1}}\right)$, $G_{m}=\left(g_{m1},g_{m2},\ldots ,g_{mk_{2}}\right)$, $G_{a}\left(g_{a1},g_{a2},\ldots ,g_{ak_{3}}\right)$ were obtained, where $k_{1}$, $k_{2}$ and $k_{3}$ represent the number of participants in NC, MCI, and AD, respectively. We calculated the average correlation matrix of each group, and each average correlation matrix was threshold with the same θ=0.15 to construct the corresponding real brain networks, i.e., $\bar{G_{n}}$ for NC, $\bar{G_{m}}$ for MCI and $\bar{G_{a}}$ for AD. The real brain network in each group consisted of a constant number of nodes, |V|=90. The connection number of $\bar{G_{n}}$, $\bar{G_{m}}$ and $\bar{G_{a}}$ were represented by $|E_{n}|$, $|E_{m}|$ and $|E_{a}|$, respectively. In current work, we studied the evolution process of AD networks from two stages, i.e., the stage from NC to MCI and the stage from NC to AD. By comparing the elements in the binary graphs $\bar{G_{n}}$, $\bar{G_{m}}$ and $\bar{G_{a}}$, we obtained a constant number $\alpha_{1}$ of connections that need to be added; another constant number $\beta_{1}$ of connections that need to be deleted in the stage from NC to MCI; and $\alpha_{2}$ connections to be added and $\beta_{2}$ connections to be deleted from NC to AD. Here, $\alpha_{1}\leq \beta_{1}$ and $\alpha_{2}\leq \beta_{2}$ because a declining number of connections was found when comparing $|E_{m}|$ and $|E_{a}|$ with $|E_{n}|$ ($|E_{a}\left|\leq \right|E_{n}\left|\leq \right|E_{m}|$). It is worth mentioning that we call the real brain network $\bar{G_{n}}$ of NC the initial network in our modeling, and call $\bar{G_{m}}$ and $\bar{G_{m}}$ the real target brain networks (TN).Connection probabilities calculation: After initialization, we calculated the connection probabilities of any node pairs in $\bar{G_{n}}$ according to the proposed connection probabilities models, i.e., ECM and MINM introduced in the following subsection. Then, we sorted each node pair in line with its connection probability. The node pair with the largest connection probability and the node pair with the smallest connection probability were recorded, respectively.Evolution: Our model started to evolve from the initial graph $\bar{G_{n}}$. In each iteration, a random number was generated to decide whether to add or delete one connection in $\bar{G_{n}}$. The node pair with the largest connection probability will establish a link if its two nodes disconnected with each other. Meanwhile, the node pair with the smallest connection probability would cut off its link if there were a connection between its two nodes. It should be noted that each node must have a connection to ensure the connectivity of the synthetic network. Therefore, a new pair of nodes must be chosen according to the sorted connection probabilities, if either node’s connection number in the node pair is equal to 1 when deleting the link between them. We upgraded connection set in $\bar{G_{n}}$ at the end of this step.End of the modeling: Our model ran the above steps of connection probabilities calculation and evolution round by round. The simulation did not proceed to the end, if $\alpha_{1}$ connections were established and $\beta_{1}$ connections were deleted successfully for $\bar{G_{n}}$ in the stage from NC to MCI; or $\alpha_{2}$ connections were established and $\beta_{2}$ connections were deleted in the stage from NC to AD. Finally, we obtained two synthetic networks with the same connection size as $\bar{G_{m}}$ and $\bar{G_{a}}$, respectively.

#### 2.4.2. Connection Probabilities Models

To explore the formation mechanism of generating human brain network topologies, Vértes et al. proposed the Economical Clustering Model (ECM) [26]. Their experiment results show that both the **topological similarity** of common neighbors (CN) and the **Euclidean distance similarity** between two brain regions are treated as essential impactors in brain networks modeling. The connection probability of ECM is defined in the following function:Connection probability of ECM: The more common neighbors (CN) that node *u* and node *v* have, the higher topological similarity to one another [48]. The number of common neighbors between node *u* and node *v* can be described by
(1)$$S_{\left(u,v\right)}^{CN}=\left|\Gamma \left(u\right)⋂\Gamma \left(v\right)\right|$$
where Γ(u) and Γ(v) represent the set of neighboring nodes of node *u* and node *v*, respectively. Scaling $S_{\left(u,v\right)}^{CN}$ by the Euclidean distance similarity E(u,v) between two brain regions, the ECN connection probability between node *u* and node *v*, i.e., the probability that *u* prefers to build the connection with *v*, is given by
(2)$$P\left(u,v\right)={S_{\left(u,v\right)}^{CN}}^{\gamma }\cdot E{\left(u,v\right)}^{-\eta }.$$

In this expression, P(u,v) is the connection probability of ECM. $S_{\left(u,v\right)}^{CN}$ is the contribution of the topological similarity computed by CN. The other term, E(u,v), is the contribution of the Euclidean distance similarity which represents the anatomical distance between nodes *u* and *v*. γ and η represent the parameters of topological similarity and anatomical distance penalization, respectively. Although ECM can construct networks similar to the real target brain networks on several essential topological features, there are still two major problems that have been pointed out.

One problem is that ECM is devoted to calculating the topological similarity between node *u* and node *v* only according to the number of their common neighbors; it does not consider the individual characteristics of node *u* and node *v*. In fact, the individual topological characteristics of node *u* and node *v* are paramount for the formation of a connection between them. Additionally, the other problem is that ECM gives each common neighbor the same score to the topological similarity. However, various common neighbors may have different topological characteristics, e.g., degree and clustering coefficient. Therefore, they make a different contribution to establishing the connection. In this study, we propose a novel brain network model named MINM from the perspective of mutual information to solve above issues. Mutual information provides a measure of the statistical relationship between two random variables. More specifically, it is a measure of the reduction in uncertainty about one random variable given knowledge of another [32]. MINM adopts not only the individual features of node *u* and node *v*, but also the topological-based mutual information of their common neighbors. The topological-based mutual information is used to distinguish the different contributions of the common neighbors of the node pair (u,v) in calculating the existence probability of one connection. Moreover, higher mutual information indicates a substantial reduction in the uncertainty of the formation of a connection; otherwise, lower mutual information means a smaller probability of the existence of one connection. The question we investigate is whether the mutual information between node *u* and node *v* can be helpful for the modeling of brain networks during the evolution process from NC to AD. Next, we introduce the definitions of self-information and mutual information, and then, we give a precise definition of our proposed connection probability of MINM.

**Definition** **1.**
*(Self-information) The Self-information of a random variable is a function that concentrates on quantifying the information involved in the value of a random variable. Given a random variable X with a probability distribution P(X), where X takes on values in a set $X=\{x_{1},x_{2},\cdots ,x_{n}\}$, the self-information of the random variable can be expressed by*
(3)$$I\left(X\right)=-\sum_{x\in X}P\left(x\right)logP\left(x\right).$$


**Definition** **2.***(Mutual information) Mutual information is a quantity that measures the number of messages that can be acquired about one random variable by observing another. Formally, the mutual information of two random variables X and Y, whose joint distribution is given by P(X,Y), can be defined as*(4)$$\begin{array}{cc} I(X;Y) & =\sum_{x\in X}\sum_{y\in Y}p\left(x,y\right)log\frac{p(x,y)}{p(x)p(y)}=\sum_{x\in X}\sum_{y\in Y}p\left(x,y\right)log\frac{p(y)p(x|y)}{p(x)p(y)}\\ & =\sum_{x\in X}\sum_{y\in Y}p\left(x,y\right)log\frac{p(x|y)}{p(x)}=\sum_{x\in X}\sum_{y\in Y}p\left(x,y\right)logp(x|y)-\sum_{x\in X}\sum_{y\in Y}p\left(x,y\right)logp(x)\\ & =-\sum_{x\in X}p\left(x\right)logp\left(x\right)-\left[-\sum_{x\in X}\sum_{y\in Y}p\left(x,y\right)logp(x|y)\right]\\ & =I(X)-I(X|Y) \end{array}$$
where P(X) and P(Y) represent the *marginal distributions* of *X* and *Y*, respectively. I(X;Y)=0 if and only if *X* and *Y* are independent.
Connection probability of MINM: Given a pair of node (u,v), whose common neighbors can be represented by $\omega_{u,v}=\Gamma \left(u\right)⋂\Gamma \left(v\right)$, the connection probability of MINM between them can be given by
(5)$$P\left(u,v\right)={S_{\left(u,v\right)}^{MI}}^{\gamma }\cdot E{\left(u,v\right)}^{-\eta }.$$
where $S_{u,v}^{MI}$ is the topological similarity between node *u* and node *v*, which represents the topology-based mutual information between them. E(u,v) is the Euclidean distance similarity, which is similar with ECM. $S_{u,v}^{MI}$ can be given by
(6)$$S_{u,v}^{MI}=-I\left(L_{u,v}^{1}|\omega_{u,v}\right)$$
where $I(L_{u,v}^{1}|\omega_{u,v})$ denotes the conditional self-information of an event that there is a connection between node *u* and node *v*, whose common neighbors are known as $\omega_{u,v}$. From Equation (Equation 6), we can know that the smaller $I(L_{u,v}^{1}|\omega_{u,v})$ is, the higher topological similarity $S_{u,v}^{MI}$ is, and this indicates that a larger probability for node *u* and node *v* to establish one connection between them.

Next, we give a comprehensive illustration about how to calculate $S_{u,v}^{MI}$. According to the definition of Mutual Information in Equation (Equation 4), the topological similarity of $S_{u,v}^{MI}$ can be given by
(7)$$S_{u,v}^{MI}=-I\left(L_{u,v}^{1}|\omega_{u,u}\right)=I\left(L_{u,v}^{1};\omega_{u,v}\right)-I\left(L_{u,v}^{1}\right)$$
where $I(L_{u,v}^{1};\omega_{u,v})$ denotes the mutual information between two events $L_{u,v}^{1}$ and $\omega_{u,v}$. $L_{u,v}^{1}$ describes the event that there is one link between node *u* and node *v*, and $\omega_{u,v}$ represents the event that the common neighbors between node *u* and node *v* are observed. $I(L_{u,v}^{1};\omega_{u,v})$ evaluates the increment of the probability for the formation of one connection between node *u* and node *v* by knowing the information of their common neighbors. $I(L_{u,v}^{1})$ represents the self-information of the event that node *u* and node *v* are connected. According to the definition in Equation (Equation 3), $I(L_{u,v}^{1})$ can be obtained by calculating the probability of $p(L_{u,v}^{1})$. In this work, we assumed that the common neighbors in $\omega_{u,v}$ are independent with each other, and thus,
(8)$$I\left(L_{u,v}^{1};\omega_{u,v}\right)=\sum_{z\in \omega_{u,v}}I(L_{u,v}^{1};z).$$

Given one common neighbor *z* in $\omega_{u,v}$, $I(L_{u,v}^{1};z)$ can be estimated by calculating the average mutual information of all pairs of nodes that have node *z* as their neighbor in common.
(9)$$\begin{array}{cc} I(L_{u,v}^{1};z) & =\frac{1}{\left|\Gamma (z)|(|\Gamma (z)|-1\right)}\sum_{m,n\in \Gamma (z)}I(L_{m,n}^{1};z),\forall m\neq n,\\ & =\frac{1}{\left|\Gamma (z)|(|\Gamma (z)|-1\right)}\sum_{m,n\in \Gamma (z)}\left\{I\left(L_{m,n}^{1}\right)-I\left(L_{m,n}^{1}|z\right)\right\},\forall m\neq n \end{array}$$
where Γ(z) is the neighbor set of *z*; $I(L_{m,n}^{1})$ represents the self-information of an event that node *m* and node *n* are connected; and $I(L_{m,n}^{1}|z)$ is used to describe the conditional self-information of the event that node *m* and node *n* are connected on the condition that node *z* is known as one of their common neighbors. Here, we can get $I(L_{m,n}^{1})$ through calculating $p(L_{m,n}^{1})$. $p(L_{m,n}^{1})$ can be given by
(10)$$\begin{array}{c} p\left(L_{m,n}^{1}\right)=1-p\left(L_{m,n}^{0}\right)=1-\prod_{i=1}^{k_{n}}\frac{\left|E\right|-k_{m}-i+1}{|E|-i+1}=1-\frac{C_{\left|E\right|-k_{m}}^{k_{n}}}{C_{\left|E\right|}^{k_{n}}} \end{array}$$
where $L_{m,n}^{0}$ represents the event that node *u* and node *v* are disconnected with each other; |E| represents the existing number of connections in the network; and $k_{m}$ and $k_{n}$ denote the connection number of node *m* and node *n*, respectively.

Then, we give an explanation of the calculation of $I(L_{m,n}^{1}|z)$, which is described in Equation (Equation 9). According to the definition of self-information, to obtain $I(L_{m,n}^{1}|z)$ we need to calculate the $p(L_{m,n}^{1}|z)$. In the current work, $p(L_{m,n}^{1}|z)$ is equal to the clustering coefficient of node *z*
(11)$$p\left(L_{m,n}^{1}|z\right)=\frac{N_{\wedge z}}{N_{\wedge z}+N_{\vee z}},$$
where $N_{\wedge z}$ and $N_{\vee z}$ represent the connected and disconnected link number in Γ(z), respectively. Obviously, $N_{\wedge z}+N_{\vee z}=|\Gamma \left(z\right)|(|\Gamma \left(z\right)|-1)/2$, which represents the total connections that could possibly exist within Γ(z). Then, $p(L_{m,n}^{1}|z)$ can be described by
(12)$$p\left(L_{m,n}^{1}|z\right)=\frac{2|E_{z}|}{\left|\Gamma (z)|(|\Gamma (z)|-1\right)},$$
where Γ(z) represents neighbors set of node *z*, $|E_{z}|$ denotes the existing link number in |Γ(z)|. Therefore, $I(L_{u,v}^{1};z)$ is given by
(13)$$\begin{array}{cc} I(L_{u,v}^{1};z) & =\frac{1}{\left|\Gamma (z)|(|\Gamma (z)|-1\right)}\sum_{m,n\in \Gamma (z)}I(L_{m,n}^{1}-I\left(L_{m,n}^{1}|z\right)),\forall m\neq n\\ & =\frac{1}{\left|\Gamma (z)|(|\Gamma (z)|-1\right)}\sum_{m,n\in \Gamma (z)}\left\{-logp\left(L_{m,n}^{1}\right)-\left(-logp\left(L_{m,n}^{1}|z\right)\right)\right\},\forall m\neq n\\ & =\frac{1}{\left|\Gamma (z)|(|\Gamma (z)|-1\right)}\sum_{m,n\in \Gamma (z)}\left\{log\frac{C_{\left|E\right|}^{k_{n}}}{C_{\left|E\right|}^{k_{n}}-C_{\left|E\right|-k_{m}}^{k_{n}}}+log\frac{2|E_{z}|}{\left|\Gamma (z)|(|\Gamma (z)|-1\right)}\right\},\forall m\neq n. \end{array}$$

Finally, substituting Equations (8), (10) and (13) back into Equation (Equation 7), we can get the topological similarity of $S_{u,v}^{MI}$.
(14)$$\begin{array}{cc} S_{u,v}^{MI}= & \sum_{z\in \omega_{u,v}}\left\{\frac{1}{\left|\Gamma (z)|(|\Gamma (z)|-1\right)}\sum_{m,n\in \Gamma (z)}\left\{log\frac{C_{\left|E\right|}^{k_{n}}}{C_{\left|E\right|}^{k_{n}}-C_{\left|E\right|-k_{m}}^{k_{n}}}+log\frac{2|E_{z}|}{\left|\Gamma (z)|(|\Gamma (z)|-1\right)}\right\}\right\}\\ & -log\frac{C_{\left|E\right|}^{k_{n}}}{C_{\left|E\right|}^{k_{n}}-C_{\left|E\right|-k_{m}}^{k_{n}}},\forall m\neq n. \end{array}$$

As we can see from Equations (7) and (14), both the mutual information $\sum_{z\in \omega_{u,v}}I\left(L_{u,v}^{1};z\right)$ and the self-information $I(L_{u,v}^{1})$ are used to define the topological similarity between node *u* and node *v*. Thus, two different node pairs (u,v) who have the same number of common neighbors may get various topological similarity.

### 2.5. Evaluation of Synthetic Networks

To evaluate the performance of our proposed model, we defined a similarity index SI function as described in [26,27], by comparing different topological properties between synthetic networks and real target brain networks (TN), i.e., the real brain networks of MCI and AD. Moreover, to make it a more convincing and reasonable evaluation, the most important network properties including clustering coefficient, local efficiency, modularity, characteristic path length, global efficiency, and transitivity are chosen. The detailed description of these features is shown in Table 2, and they can reflect the performance of one network from different aspects. The definition of SI is given by
(15)$$SI=\frac{1}{\xi_{C}+\xi_{Eloc}+\xi_{M}+\xi_{L}+\xi_{Eglob}+\xi_{T}}.$$
where $\xi_{C}$ is the relative error of clustering coefficient between the synthetic network and real target brain network (TN); $\xi_{Eloc}$ and $\xi_{Eglob}$ represent the relative errors of local efficiency and global efficiency, respectively. Similarly, $\xi_{M}$, $\xi_{L}$, and $\xi_{T}$ are the relative errors of modularity, the characteristic path length, and transitivity, respectively. A smaller value of SI indicates that the synthetic networks are more similar to the real target brain networks of MCI or AD. Obviously, we can get different values of SI with various γ and η. To find the optimal synthetic networks that overall most closely approximate the real target group, simulated annealing (SA) on SI is used. Moreover, the optimal γ and η in the parameter space that minimizes SI are recorded.

## 3. Results

In this study, we propose one novel brain network model named MINM for a better understanding of the connection mechanism of AD brain networks from two stages, i.e., the stage from NC to MCI, and the stage from NC to AD. Both topological-based mutual information and the anatomical distance are taken into account in MINM. We give detailed descriptions of the modeling results in the two stages, respectively, in the following subsections.

### 3.1. Topological Differences in Brain Networks of NC, MCI and AD

Before introducing the modeling results, we discuss the topological differences among brain networks of NC, MCI, and AD. Figure 1 shows the changes of the six topological features among the real observed networks of NC, MCI, and AD groups. The results indicate that noticeable topological changes are found in the networks of these three groups. Specifically, clustering coefficient, local efficiency, global efficiency, and transitivity of brain networks decrease gradually from NC, via MCI to AD, whereas characteristic path length and modularity increase progressively. The analysis of these topological differences can help us explain how connections change in brain networks, and it is valuable for proposing a more reasonable simulation model.

### 3.2. Network Modeling of the Stage from NC to MCI

In this subsection, we evaluate the performance of our proposed model MINM and five other well-known models in the stage from NC to MCI. The compared models are similar with MINM except for the definitions of the topological similarities. Table 3 makes a detailed description of the topological similarities in the compared models. In addition, we also perform a random model for comparison. In the random model, we selected two disconnected nodes to establish one connection and two connected nodes to delete one connection, randomly.

Table 4 shows the SI of different models with the best-fitting parameters λ and η. We found that MINM outperformed the other models with the largest SI=3.9683 and the optimal parameters λ=0.4,η=2.0, followed by RA (SI=2.4358,λ=0.4,η=2.0), PA (SI=2.4010,λ=0.4,η=1.8) and JC (SI=2.3714,λ=0.2,η=0.2). The results indicated that MINM considering both network topology-based mutual information and anatomical distance could generate synthetic networks capturing all of the key topological features of real target brain networks. Specifically, MINM minimized the difference between the synthetic network and the real brain network on properties of local efficiency ($\xi_{Eloc}=0.0077$), global efficiency ($\xi_{Eglob}=0.0292$), the characteristic length ($\xi_{L}=0.0598$), and transitivity ($\xi_{T}=0.0010$). Therefore, MINM was the best model in simulating the underlying mechanism that causes the connection changes in brain networks in the stage from NC to MCI. Moreover, we found that PA generated synthetic networks most similar with the real brain network of MCI on properties of clustering coefficient, but it failed to match real MCI networks well on transitivity ($\xi_{T}=0.0480$). RA generated synthetic networks most similar with the real brain network of MCI on properties of modularity, but mismatched real MCI networks on properties of clustering coefficient ($\xi_{C}=0.0975$) and local efficiency ($\xi_{Eloc}=0.0638$). In contrast, the results also indicated that the SI of ECM was the smallest among the six topology-based models. The performance of ECM was limited primarily by mismatches in characteristic path length ($\xi_{L}=0.1851$) and global efficiency ($\xi_{Eglob}=0.1413$). Additionally, we also compared the six topology-based models with the Random model. As shown in Table 4, we found that the SI of random model had the lowest SI=1.0912.

Furthermore, to make a detailed illustration of the above results, we also present the values of the six topological properties in both synthetic networks generated by the topology-based models and the real target brain network (TN) of MCI in Figure 2. In the figure, the black dot lines indicate the property values of TN, and the red dot lines describe the property values of the best models which minimize the relative errors.

### 3.3. Network Modeling of the Stage from NC to AD

We studied the evolution process of network connections in the stage from NC to MCI. SI was again used to evaluate the performance of our proposed model MINM and five other known topology-based models. A random model was also performed for comparison. By comparing SI and the relative errors of different topological properties, we estimated the connection mechanism that causes the changes of connections in AD patients.

Table 5 gives a detailed illustration of the largest SI for the six topology-based models and the random model. The corresponding best-fitting parameters λ and η and the relative errors on the six topological properties are listed in the table. We found good correspondence between the synthetic network constructed by MINM and the real target brain network of AD on all of the key topological properties of global efficiency ($\xi_{Eglob}=0.0303$), the characteristic length ($\xi_{L}=0.0253$), and transitivity ($\xi_{T}=0.0368$). Overall, the results confirmed that our proposed model MINM achieved the largest SI=3.9231 with the optimal parameters λ=0.2,η=1.8, which meant the corresponding synthetic network was significantly more AD-like than either of the other topological models previously considered. Therefore, MINM was the best model for anticipating the connection mechanism of the real brain network in the stage from NC to AD. In addition, the SI of PA was equal to 2.4358, with the optimal parameters λ=0.4,η=2.0, preceded only by MINM. The SI and the optimal parameters of AA and RA were SI=2.4010,λ=0.4,η=1.8 and SI=2.3714,λ=0.2,η=0.2, respectively, followed by PA. Another finding was that the synthetic network generated by PA had clustering coefficient and modularity that exactly matched with that of the real brain network of AD. $\xi_{Eloc}$ of the synthetic network generated by RA was the closest to the real target brain network. However, the synthetic network failed to match the real network well regarding modularity ($\xi_{M}=0.1370$) and the characteristic path length ($\xi_{L}=0.0732$). Furthermore, the results of the current study also revealed that the performance of JC was not as clustering or local efficient as the real brain network of AD. ECM had the worst performance in simulating the global efficiency ($\xi_{Eglob}=0.1413$), and it obtained the smallest SI among the six topology-based models. Finally, the comparison results showed that all the six topology-based models outperformed the random model, significantly.

Detailed information about the six topological properties of the synthetic networks and the real target brain network (TN) of AD are shown in Figure 3.

### 3.4. Degree Distribution

In this subsection, we explore the degree distribution of the synthetic networks. In the study of graph theory, the degree of a node indicates the connections number it has to other nodes. The node degree describes the characteristic of the network from the view of local structures. The degree distribution is the probability distribution of these degrees over the whole network. It makes an evaluation of the structural and dynamical properties in the network from a global view. Therefore, it is necessary to make a detailed illustration of the degree distribution of the synthetic networks in the current study. Figure 4 shows the fitting results of the cumulative degree distribution of both the real target brain networks (TN) of MCI and AD, and synthetic networks generated by the topology-based models in the simulation of two specified stages, i.e., the stage from NC to MCI and the stage from NC to AD.

Figure 4a demonstrates the fitting results of the cumulative degree distribution for synthetic networks generated in the stage from NC to MCI, and the real MCI network in a log-log plot. Figure 4b shows the fitting results of the cumulative degree distribution for synthetic networks generated in the stage from NC to AD, and the real AD network. Both results show that the generated topologies of MINM were approximately the same as the real target brain networks (TN) of MCI and AD. Additionally, the results indicate that a significant difference existed between the synthetic network generated by the random model and the real target brain networks (TN).

### 3.5. Topological Properties of the Synthetic Networks Generated by MINM with Different λ and η

We subsequently investigated the influence of the variety of parameters λ and η on the topological properties in the modeling stage from NC to AD. Figure 5a–c describes the changes of the topological properties with η, when λ=0.2 is a constant. An interesting finding is that the clustering coefficient, local efficiency, characteristic path length, and transitivity increased with a gradual increase of η. These properties obtained the largest values when η=1.8, which is consistent with the observation from the results in Table 5. It is worth mentioning that modularity and global efficiency did not change greatly with the increase of η, confirming that the anatomical distance makes little influence on these two features. Additionally, we also investigated the topological properties of the networks by varying λ with η=1.8. As can be seen in Figure 5d–f, we observed a decreasing trend in clustering coefficient, local efficiency, characteristic path length, and transitivity with the increase of λ from 0.2 to 2.0. Especially, these properties obtained the largest values when λ=0.2, which indicates a good agreement with the results in Table 5. It is interesting to note that both the modularity and global efficiency change little with the increase of λ, confirming that the topological similarity has little influence on these two features.

### 3.6. Connections Deleted in the Early Stage from NC to AD

As described in the above results, we proved that our proposed MINM could achieve good performance in the progression simulation of AD. In this subsection, we show how a normal brain network converts to an AD brain network step-by-step by displaying the simulated progress of MINM. Specifically, we explore which connections are deleted in the early stage of AD. Table 6 records the connections disrupted between particular brain regions. (These brain regions are defined according to the Automated anatomical labeling (AAL) template. The detailed number and abbreviation of each region can be accessed at the following link, http://neuro.imm.dtu.dk/wiki/Automated_Anatomical_Labeling). As can be seen in Table 6, the connections linked to brain regions such as Region 8 (Frontal_Mid_R), Region 85 (Temporal_Mid_L), Region 51 (Occipital_Mid_L), Region 53 (Occipital_Inf_L) and Region 54 (Occipital_Inf_R) disrupt frequently in the early stage of transition. The finding of these early-loss connections may be helpful for early detection of AD in clinical.

In addition, we also describe the topological properties changes simulated by MINM in the progress of deleting a different number of connections. As shown in Table 7, we find that the properties such as clustering coefficient, local efficiency, global efficiency, and transitivity decrease progressively along with the increasing of the deleted connection number. Moreover, characteristic path length and modularity increase gradually. These variations are consistent with the findings of Figure 1.

## 4. Discussion

In this study, we investigated several brain network models to explore the fundamental reason that results in the alteration of connections in real AD brain networks. We showed that the previous models, which consider either node feature, i.e., node degree, or shared nearest neighbors in common, could not satisfactorily account for all the topological properties of real target brain networks. We demonstrated that the addition of the topological-based mutual information to the model, favoring additional formation of connections between two nodes, could markedly improve the simulation of realistic brain network properties. Notably, our proposed model MINM provided a good account of network properties regarding local efficiency, global efficiency, the characteristic path length, and transitivity. MINM also provided an excellent fit to the degree distribution in both simulation stages of MCI and AD. Therefore, we confirm that MINM is a promising method for understanding the underlying mechanism that shapes the organization of brain networks during the progression of AD.

We intensively studied the topology differences between functional brain networks of NC and AD, and significant alterations of topological properties were found, i.e., the increase of characteristic path length and the decrease of network efficiency. Specifically, the increasing of the characteristic path length indicates a long average time in dealing with the information transmission among brain areas; and the dropping down of both local efficiency and global efficiency implies the reduction of the ability in data processing. The alterations of these properties are related to the functional decline of cognition in AD patients, which is consistent with results in the literature [43,44]. These previous works show that the abnormal alterations of networks topologies play vital roles in the influence of cognitive functions such as attention, working memory and executive control of human brains during the neuro-development. The disorder of network organizations will lead to the deficit of cognition and result in the emergence of AD. Therefore, better understanding the topological characteristics would be helpful for us to know the relationship between the functions and the architectures of AD brain networks.

In the current work, we have studied the changes of network properties for a further investigation of the fundamental rule that causes the formation of AD network topologies. Prior models, such as ECM, AA, and PA, estimate the connection mechanism of the brain networks formation by emphasizing the importance of increasing connections among brain regions [26,27]. Although the authors have confirmed that synthetic networks generated by these models can well match several properties of the real target brain networks, such an effort is accompanied by much debate in recent years in the simulation of the progression of AD. One obvious limitation of previous models is that they ignore the fact that the alterations in AD brain networks are caused by both decreasing and increasing connections between brain regions [43,44]. Consequently, previous models cannot successfully identify the potential mechanism that results in the variation of network topologies in the progression from NC and MCI, or NC and AD. It is worth mentioning that our proposed model MINM takes both decrease and increase of connections among brain regions into account, and this consideration is suitable for the actual situation of AD. Thus, it obtains good performance in the modeling of AD brain networks.

It is also clear that mutual information has been extensively investigated in human brain research. The previous study has demonstrated that mutual information is associated with the cognitive functions of human brain networks [37,38,40]. In addition, much work has suggested that mutual information makes a crucial influence on information transmission in brain networks [36]. Therefore, topological-based mutual information is considered an essential factor in MINM to make the modeling more efficient. MINM can appropriately distinguish the connection probabilities between any two pairs of nodes by considering the topological-based mutual information. This consideration may make it much reasonable for capturing the potential connection mechanism that shapes the topologies of AD brain networks. Experimental results are shown in Table 4 and Table 5. Based on our findings, it can be concluded that our proposed topological-based mutual information model MINM can make a complete understanding of the connection mechanism that causes the formation of AD networks. Furthermore, we know that our proposed model MINM can generate networks with a better simulation in network properties of global efficiency and transitivity in both modeling stage from NC to MCI and from NC to AD. It means that the synthetic network has the same ability as the real ones in dealing with the global information transformation.

We discuss several competitive models that consider both topological similarity and distance penalization in this work. We make a further analysis of the influence of different topological structures on the reconstruction of AD brain networks. In the ECM model, two nodes get higher connection probability if they share more nearest neighbors in common. The PA model considers node degree of one node when calculating the connection probability, and the node with a large degree, i.e., the hub node, has more opportunity to establish connections with others. While AA, RA, and JC are similar to ECM, they all take the number of common neighbors into consideration when calculating the connection probability between two nodes. Especially, both AA and RA take the individual characteristic of the node, i.e., node degree into account besides the common neighbors. JC defines the topological similarity for connection establishment with the consideration of not only the number of the common neighbors of two nodes, but also their union neighbor set. As can be seen in Table 4 and Table 5, all the topological-based models generated synthetic networks much more similar to the real target brain networks than the random model. Moreover, we also found that models taking different topological structures into account generated various network topologies that present their characteristics. Additionally, through comparing SI of MINM to those of other topological-based models, we confirmed that MINM achieved the best performance in generating synthetic networks with properties most resemblance to those of the real AD brain network, with the best fitting parameters λ and η estimated by simulated annealing (SA). We devote our effort to propose an explicit simulation model for a complete understanding of the connection mechanism that results in the network topologies of AD. In addition, our results reveal that the formation of the AD topologies is not random; it follows an appropriate rule. Our findings draw attention to the critical requirement for a comprehensive understanding of the relationship between the connection mechanism and the functions of brain networks.

## 5. Conclusions

We have investigated how topological-based mutual information and Euclidean distance are adopted in the simulation of brain network topologies with AD. We also concentrate on how connections are established or deleted among different brain regions. Our ambition is to uncover the fundamental connection mechanism that facilitates the alterations of brain networks in the progression from NC to AD. We demonstrate that adding the mutual information into our model can promote the modeling performance in this progression. Successful models of AD brain networks have been instrumental in understanding how structural brain organizations affect the ability of cognition. Our work has opened new avenues toward the diagnosis and treatment of AD.

## Figures and Tables

**Figure 1 entropy-21-00300-f001:**
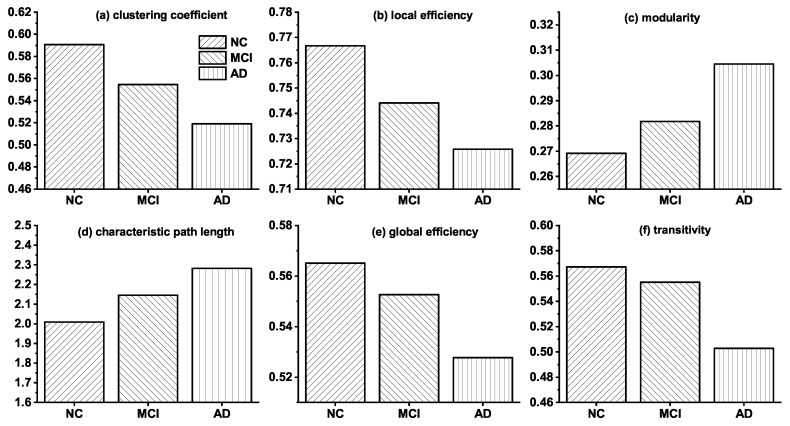
Topological differences among the real brain networks of NC, MCI and AD. NC represents the real brain network of normal control (NC) group; MCI and AD are the real brain networks of Mild cognitive impairment (MCI) and Alzheimer’s disease (AD) group, respectively.

**Figure 2 entropy-21-00300-f002:**
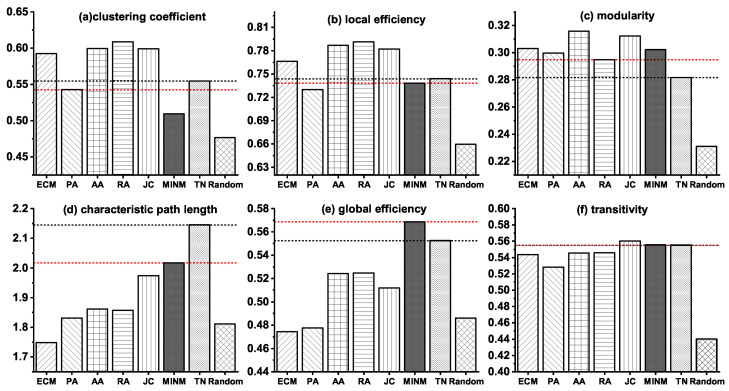
Topological properties of the synthetic brain networks generated by various models and the real target brain network (TN) of MCI.

**Figure 3 entropy-21-00300-f003:**
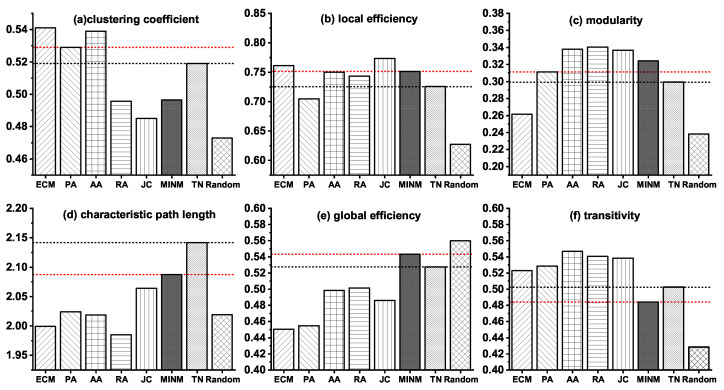
Topological properties of the synthetic brain networks generated by various models and the real target brain network (TN) of AD.

**Figure 4 entropy-21-00300-f004:**
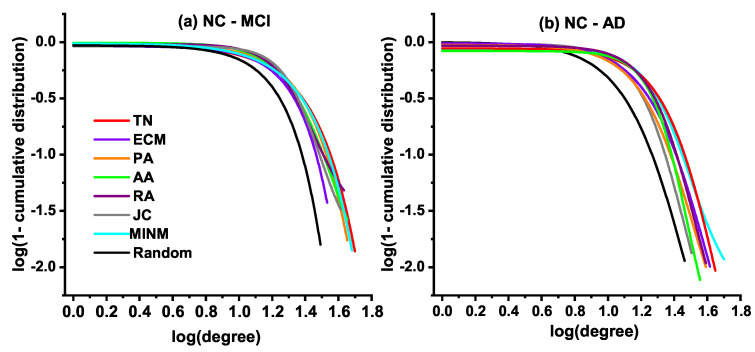
Degree distributions of the real target brain networks (TN) and the synthetic brain networks.

**Figure 5 entropy-21-00300-f005:**
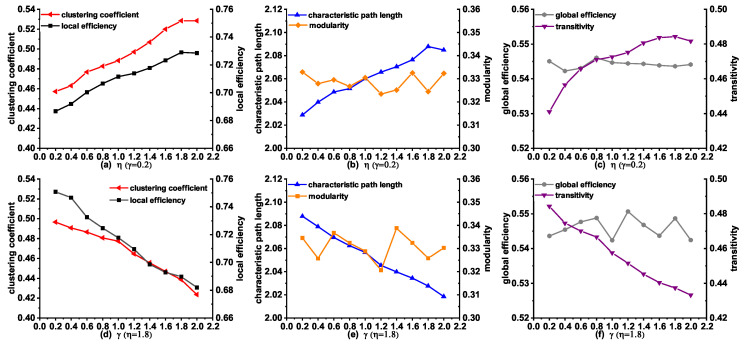
The changes of topological properties of the synthetic networks generated by MINM with different λ and η in the stage from NC to AD.

**Table 1 entropy-21-00300-t001:** Demographic and clinical characteristics of the participants in Normal controls (NC), Mild cognitive impairment (MCI) and Alzheimer’s disease (AD) groups.

	NC	MCI	AD
Number	62	45	40
Gender (Male/Female)	27/35	20/25	21/19
Age	73.95 ± 4.83	74.38 ± 4.92	74.86 ± 5.52
MMSE score	28.72 ± 1.06	27.68 ± 1.86	22.36 ± 2.77
CDR score	0.00 ± 0.00	0.51 ± 0.17	0.93 ± 0.16

Values of Age, MMSE score and CDR score are expressed as the mean ± SD (standard deviation). MMSE: Mini-Mental State Examination; CDR: Clinical Dementia Rating. Significant differences were noted in MMSE scores between any two groups (*p* < 0.05, the *p*-value was obtained by two sample *t*-test).

**Table 2 entropy-21-00300-t002:** Description of topological properties in complex networks.

Property Name	Symbol	Description
Clustering coefficient	C	It is a measure of the number of triangles in a graph.
Local efficiency	Eloc	It is a measure to quantify the efficiency of local information transmission.
Global efficiency	Eglob	It is a measure to quantify the efficiency of global information transmission.
Characteristic path length	L	L is the average shortest path length between all node pairs in the network.
Modularity	M	It is used to detect the strength of the division of a network into communities.
Transitivity	T	It measures the probability that the adjacent nodes of a node are connected.
Degree	k	It indicates the number of links connecting with a node.

**Table 3 entropy-21-00300-t003:** The definitions of topological similarities in the compared models.

Models	Abbreviation	Mathematical
Preferential Attachment [49]	PA	$S_{u,v}^{PA}=\left|\Gamma \left(u\right)\right|\times \left|\Gamma \left(v\right)\right|$
Jaccard [50]	JC	$S_{u,v}^{JC}=\frac{\left|\Gamma (u)⋂\Gamma (v)\right|}{\left|\Gamma (u)⋃\Gamma (v)\right|}$
Adamic–Adar [51]	AA	$S_{u,v}^{AA}=\sum_{\xi \in |\Gamma (u)⋂\Gamma (v)|}\frac{1}{log\left|\Gamma (\xi )\right|}$
Resource Allocation [52]	RA	$S_{u,v}^{RA}=\sum_{\xi \in |\Gamma (u)⋂\Gamma (v)|}\frac{1}{\left|\Gamma (\xi )\right|}$

**Table 4 entropy-21-00300-t004:** The optimal SI-value of different brain network models for networks modeling in the stage from NC to MCI. $\xi_{C}$ is the relative error of clustering coefficient between synthetic networks and the real target brain network (TN); $\xi_{Eloc}$ and $\xi_{Eglob}$ represent the relative errors of local efficiency and global efficiency; and $\xi_{M}$, $\xi_{L}$, and $\xi_{T}$ are the relative errors of modularity, the characteristic path length, and transitivity, respectively. A larger value of SI indicates that the model could generate synthetic networks with properties more similar to the real target brain network of MCI.

Models	λ	η	$\xi_{C}$	$\xi_{\mathrm{Eloc}}$	$\xi_{M}$	$\xi_{L}$	$\xi_{\mathrm{Eglob}}$	$\xi_{T}$	*SI*
ECM	0.2	1.6	0.0687	0.0302	0.0713	0.1851	0.1413	0.0205	1.9339
PA	0.2	1.8	**0.0213**	0.0187	0.0639	0.1464	0.1356	0.0480	2.4010
AA	0.2	1.4	0.0811	0.0578	0.1210	0.1324	0.0514	0.0173	2.1692
RA	0.4	2.0	0.0975	0.0638	**0.0465**	0.1345	0.0502	0.0164	2.4358
JC	0.2	0.2	0.0844	0.0515	0.1082	0.0861	0.0789	0.0126	2.3714
MINM	0.4	2.0	0.0816	**0.0077**	0.0727	**0.0598**	**0.0292**	**0.0010**	**3.9683**
Random	–	–	0.1406	0.1132	0.1796	0.1558	0.1204	0.2068	**1.0912**

**Table 5 entropy-21-00300-t005:** The optimal SI-value of different brain network models for networks modeling in the stage from NC to AD. $\xi_{C}$ represents the relative error in clustering coefficient between synthetic networks and the real target brain network (TN); $\xi_{Eloc}$ represents the relative error in local efficiency; $\xi_{M}$ is the relative error in modularity; $\xi_{L}$ is the relative error in the characteristic path length; $\xi_{Eglob}$ is the relative error in global efficiency; and $\xi_{T}$ is the relative error in transitivity.

Models	λ	η	$\xi_{C}$	$\xi_{\mathrm{Eloc}}$	$\xi_{M}$	$\xi_{L}$	$\xi_{\mathrm{Eglob}}$	$\xi_{T}$	*SI*
CN	0.4	1.2	0.0426	0.0491	0.1263	0.0665	0.1460	0.0419	2.1169
PA	0.2	1.6	**0.0194**	0.0289	**0.0395**	0.0551	0.1377	0.0515	3.0111
AA	0.2	1.6	0.0386	0.0339	0.1291	0.0576	0.0551	0.0881	2.4851
RA	0.2	2.0	0.0448	**0.0245**	0.1370	0.0732	0.0496	0.0758	2.4697
JC	0.4	0.4	0.0653	0.0661	0.1249	0.0364	0.0786	0.0709	2.2614
MINM	0.2	1.8	0.0432	0.0358	0.0835	**0.0253**	**0.0303**	**0.0368**	**3.9231**
Random	–	–	0.0888	0.1355	0.2304	0.0534	0.0611	0.1476	**1.3951**

**Table 6 entropy-21-00300-t006:** Detailed connections deleted in the early stage of transition. It should be noted that (85,2) in this table means there is a connection between Region 85 and Region 2. The first column “Deleted connections number = 10” records the first ten connections deleted from the NC brain network; the second and the third columns record the additional connections deleted when our model evolved to the further steps.

Deleted Connections Number = 10	Deleted Connections Number = 20	Deleted Connections Number = 30
(85,2) (53,4)	(54,7) (53,2)	(51,10) (54,23)
(77,12) (85,4)	(51,14) (49,26)	(51,23) (54,24)
(51,4) (86,85)	(52,7) (54,3)	(50,7) (49,14)
(51,8) (51,2)	(53,14) (51,24)	(53,24) (54,13)
(53,8) (85,8)	(49,8) (49,10)	(49,24) (52,9)

**Table 7 entropy-21-00300-t007:** Description of topological properties changes in the progress of deleting a different number of connections.

Deleted Connections Number	*C*	Eloc	*M*	*L*	Eglob	*T*
0	0.5906	0.7667	0.2691	2.0082	0.5651	0.5672
10	0.5829	0.7626	0.2723	2.0109	0.5639	0.5492
20	0.5784	0.7626	0.2760	2.0254	0.5616	0.5459
30	0.5764	0.7626	0.2772	2.0397	0.5575	0.5438
AD	0.5190	0.7258	0.3045	2.2821	0.5277	0.5027

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
