# Peer review of "Brain Network Modeling Based on Mutual Information and Graph Theory for Predicting the Connection Mechanism in the Progression of Alzheimer’s Disease"

_entropy, 2019, doi:10.3390/e21030300_

Reviewer 1 Report

General Comments:

Authors proposed a new model named MINM using the topology-based mutual information to model the synthetic brain network initializing from an average correlation matrix of normal controls and transition to a simulated network of Alzheimer’s disease. The topic is of interest for the certain research field, and the manuscript is well organized and written. Detailed descriptions of MINM approach are given in a step-by-step manner. Several issues are suggested to be addressed or revised before the consideration of publication.

Specific Comments:

1. Please spell out the acronym of MINM for the first time mentioned in the content. It can definitely help readers to memorize and understand the meaning of MINM.

2. Please rephrase the first sentence in abstract. Abnormal connections in brain networks do not always result in the cognitive impairment.

3. The term of “brain development” describes the maturation process that the brain producing extensive neural connections from birth to age 5. Please rephrase all the statements of incorrect usage of “development” in the manuscript, such as “the development of brain from NC to AD”.  

4. Please spell out “MMSE” for the first time mentioned in the content.

5. In Section 2.2, the signals were filtered with a passband from 0.06 to 0.11Hz. This range is pretty different from the common passband, from 0.01 to 0.1 Hz, that is widely applied for the construction of fMRI brain networks. Please clarify this issue.

6. In Section 2.4.1, the threshold of 0.15 was used to construct a binary graph for each group, however this threshold is too low because typically it takes the correlation coefficient larger than 0.3 to confirm the significance of existence (i.e., with a p-value less than 0.05). Please justify the parameter setup.

7. In Table 5, MI should be written as MINM.

8. As the authors’ statement that the connection mechanism which unravels how brain connections rewire (either by the disruptions of existed connections or the formation of new connections) from NC to AD state is essential to investigate the occurrence of AD. Displaying the progress simulated by the MINM model showing how a normal/healthy brain convert to an AD brain network step-by-step may help the elucidation. Specifically, are there certain functional connections between particular brain regions disrupt in the early stage of transition? These early-loss connections may be imaging biomarkers for the early detection of AD in clinical.

Author Response

Response to Reviewer 1 Comments

 We would like to express our great appreciation to you for comments on our paper. We hope that the correction will meet with approval. Thank you very much for your warm work earnestly.

 Detailed Response is listed in the "respond reviewer1 - MDPI.docx"

Reviewer 2 Report

Authors investigated several brain network models in the paper to explore the alteration of connection in AD patients brain network. Although the authors compared their results with previous models the results presented are not convincing. The paper is full of redundancies, for example the method section where the authors described the models can be completely removed or shortened. The results could have been presented in a much better visual way to make the results convincing. Authors have used several acronyms throughout the paper without even defining them, for example "MINM". Authors are giving the message in the paper that it is the abnormal connection (please use the better term) in brain networks which causes cognitive impairment and finally AD, this message is not true as the mechanism of AD is unclear and there is no clear path, so please remove any such misleading statement from the paper. 

Author Response

Response to Reviewer 2 Comments

 We would like to express our great appreciation to you for comments on our paper. We hope that the correction will meet with approval. Thank you very much for your warm work earnestly.

 Detailed Response is listed in the "respond reviewer2 - MDPI.docx"

Round  2

Reviewer 1 Report

Authors gave sufficient responses and revised the manuscript accordingly. Only one minor comment for the authors is to add the response (to Point 8 of my comment) into the revised manuscript. This will definitely help readers to better understand the potential application of the proposed MINM approach.

Author Response

Dear Reviewer:

Thank you for your comments in the Review Report (Round 2) concerning our manuscript entitled “Brain network modeling based on Mutual Information and Graph Theory for predicting the connection mechanism in the progression of Alzheimer's disease” (Manuscript ID:entropy-439541). We have studied comments and have made a correction. Particularly, we revised the comment (Point 8 in Round 1) carefully. The revised portion is marked with wavy lines in blue in a separate document named diffEnt2.pdf. In diffEnt2.pdf, we list all the modification and differences from the previous revised version in Round 1. The corrections in the paper and the response to your comments are as flowing:

Response to Reviewer 1 Comments in Round2

General Comments:

Authors gave sufficient responses and revised the manuscript accordingly. Only one minor comment for the authors is to add the response (to Point 8 of my comment) into the revised manuscript. This will definitely help readers to better understand the potential application of the proposed MINM approach

Response: We have revised this comment in the new manuscript, and we added a subsection 3.6 Connections deleted in the early stage from NC to AD to illustrate the connection alterations in the early stage of the transition.

Special thanks to you for your good comments.

Reviewer 2 Report

Authors have answered my comments but still the results presented are not convincing.

Author Response

Dear Reviewer:

Thank you for your comments in the Review Report (Round 2) concerning our manuscript entitled “Brain network modeling based on Mutual Information and Graph Theory for predicting the connection mechanism in the progression of Alzheimer's disease” (Manuscript ID:entropy-439541). We have carefully considered your comments on presenting better results. The revised portion is marked with wavy lines in blue in a separate document named diffEnt2.pdf. In diffEnt2.pdf, we list all the modification and differences from the previous revised version in Round 1. The corrections in the paper and the response to your comments are as flowing:

Response to Reviewer 2 Comments in Round2

General Comments:

Authors have answered my comments but still the results presented are not convincing.

Response: We have tried our best in modifying the graphs of Figure 2, Figure 3 and Figure 4 in Round 1 to make our results more clear to read. We think it is helpful for a better understanding of our manuscript.

Special thanks to you for your good comments.